# Quantitative and Qualitative Analysis of Psychosocial Factors Affecting Women’s Entrepreneurship

**DOI:** 10.3390/bs13040313

**Published:** 2023-04-05

**Authors:** David Peris-Delcampo, Antonio Núñez, Catia Miriam Costa, Marcelo Moriconi, Enrique Cantón, Alexandre Garcia-Mas

**Affiliations:** 1Department of Methodology and Basic Psychology, University of Valencia, 46010 Valencia, Spain; 2Department of Basic Psychology, University of Balearic Islands, 07122 Palma de Mallorca, Spain; 3Center for International Studies, ISCTE Instituto Universitário de Lisboa, 1649-026 Lisboa, Portugal

**Keywords:** women’s entrepreneurship, psychosocial factors, self-efficacy

## Abstract

This work aims to clarify the psychosocial variables that lead women to undertake and those that prevent them from doing so. Two studies were conducted using a mixed methodology to compensate for the inherent weaknesses of using each approach. The first study was based on the collection of quantitative data using the GloPEW questionnaire with a sample of 296 people. The second study, of a qualitative nature, was carried out through focus groups with a sample of 26 people. The results show that self-efficacy and emotional intelligence are the main factors to develop to promote entrepreneurship among women. Although the data show statistical strength, it seems necessary to expand the sample and incorporate more profiles of female entrepreneurs, for example, with different levels of training, given the complexity and variety of intervening factors.

## 1. Introduction

Gender equality is a relevant topic in the different plans of the United Nations. For example, the fifth goal of the 2030 Agenda [1] seeks to “achieve gender equality and empower all women and girls”, remembering the importance of breaking gender stereotypes and prejudices to advance this gender equality.

These stereotypes and prejudices are present in many fields and contribute to increasing the perceived barriers among women, which have been functioning, as pointed out in many studies, as a socially established “glass ceiling” that prevents women’s progression under equal conditions and opportunities [2,3,4,5].

The limitations imposed on women are more visible when moving up in positions and places of greater responsibility and power. A clear example is what happens in the business field, where, traditionally, women have encountered many barriers related to stereotypes and preconceptions. These socially generated limitations have been contributing to enhancing the psychosocial profile in women, with a lower perceived level of self-confidence, lower expectations of improvement, and a greater degree of negative emotional influence that are, at the same time, a cause and result of their difficulties to progress and undertake [6,7,8,9].

The ESTEEM project aims to provide an overview of the status of female entrepreneurship, focusing on how psychosocial factors impact entrepreneurship by gender, searching for new dynamics to stimulate women’s leadership, and implementing training on good practices. It is funded by the European Commission (Erasmus + France Program). The consortium includes partners from France (CMA AURA), Spain (APECVA), Italy (TDM 2000), and Portugal (ISCTE).

This study aims to analyze which related factors would improve the outlook on women in business. There are different aspects of the psychological factors related to the labor market in the sense of being part of the meta-competencies needed to be more effective. These factors are self-efficacy, well-being, emotional intelligence, perceived barriers, and transversal skills, such as taking the initiative or seeking support [10,11].

The selected variables are based on solid evidence of their influence in the business and work environment, from perceived social barriers and gender prejudices or stereotypes. These variables act as imitating factors and take the role of psychosocial variables and skills that, depending on whether they have been developed adequately or not, end up being elements that condition entrepreneurship and progress for women [12,13]. We expect that in the cases of some women in the analyzed sample who showed higher levels of efficiency and entrepreneurship their gender differences would be minor and they may even be comparable to men.

Finally, all the factors that can make us competent to undertake and advance in our achievements are specified in learning a series of specific entrepreneurial skills, which we need to identify and enhance, especially in the case of women. Therefore, they can carry out the actual behaviors that bring us closer to desirable equality.

Lastly, we have sought to determine how the context, the entrepreneur’s immediate environment, impacts their mental model. Thus, entrepreneurs, among others, use their cognitive capacities. They share their mental model with collaborators, partners, and the immediate environment. The situation and the environment in which the entrepreneur operates directly intervene as creative factors of entrepreneurial dynamism.

To sum up, our study focused on three major lines:Determination of the cognitive profile and organizational strategy;Modification of the cognitive profile by learning results/outcomes;Interaction with the environment for entrepreneurial dynamism.

We used a complementary quantitative/qualitative mixed methodology, which included the design of a specific evaluation instrument that could be used in both methodologies.

### 1.1. Women’s Entrepreneurship: An Evolving Concept

Several issues are pointed out in the literature as barriers to female entrepreneurship in different working sectors. However, the literature also identifies other characteristics enabling business and its success. By exploring the “entrepreneurship” concept, we conclude it has evolved and brought the gender issue into discussion. A traditional perspective of the meaning of entrepreneurship is that it is commonly understood as business ownership, business growth, and profit-making, often identified as a masculinized construct [14,15]. Researchers often criticized this view for marginalizing other interpretations, representing a “Western” idealization of business creation and success, distinguishing between genders and limiting “knowledge of entrepreneurship theory and practice” [14]. However, several authors argue that entrepreneurship is perceived more broadly and that this vision as a male construct is inadequate.

Galloway and colleagues [14] underline that even in the descriptions used to define overtly male leadership styles, “there seems to be much influence from leadership styles and characteristics commonly associated with feminized cultural markers, notably emotion, and including support, relationships and consideration, at least as characteristics of emotional intelligence”. Emotional intelligence is seen as a prerequisite for effective leadership that promotes innovation, growth, and value addition in organizations. In addition, the authors propose that leadership is also a performative concept, implying a link between entrepreneurship and leadership—and entrepreneurial leadership—that includes not just what the individual who leads is but also what they do. This perspective is based on progressive autonomy and self-determination that enable women to choose their strategies for their life and business [14,16].

The concept of women’s entrepreneurship is operationalized more broadly. Paoloni and Serafini [15] classify female entrepreneurship in four different ways, indicating that the concept can be measured as a natural and social variable (Table 1):

Only after having established female entrepreneurship as a collective variable will, it be possible, for example, to exclude that differences in value creation between male and female entrepreneurs are due to individual diversity instead of gender. In addition to the fact that they indicate leadership capacity as an essential factor for good performance in business work, another frequent denominator in these studies is the capacity for innovation [17,18,19]. According to Sarkki et al. [16], the concept of (social) innovation can be understood as a “reconstructive cycle” and defined as cyclical innovation processes involving women through civil society initiatives. It follows a process in which they question marginalizing and discriminatory practices, institutions, and cognitive structures, often relating to models of entrepreneurship (corporatism, etc.).

In our study and fundamentally following the approach of Galloway and colleagues [14], we have identified in the scientific literature the main variables that affect women’s entrepreneurship, to try to answer the question of whether they are really relevant and measurable factors, using a combined focus group strategy and the development of a scale for subsequent measurement.

On one hand, studies with psychological approaches to entrepreneurial activity help to explain why people decide or not to engage in an entrepreneurial initiative focus and the profile/personality needed to do so. In the case of women, the results highlight some specificities. Women who develop business activities indicate several factors related to their psychological characteristics, which would constitute strengths to be an entrepreneur or factors that influence the decision to be an entrepreneur [20]. The ability to spot opportunities [21], individual ambitions, objectives, and levels of self-confidence, discipline and the spirit of sacrifice [22], and personality traits such as proactivity and the ability to innovate [23], considering variables related to emotions, such as entrepreneurial passion [24], are factors that contribute to women taking up entrepreneurial activities.

On the other hand, entrepreneurship is a process in which entrepreneurs must acquire a series of technical skills to be successful. Authors refer to skills such as communication capacity; knowledge of the business area; and management skills, such as planning or decision making. They also mention personal skills, such as perceived internal control, innovation, risk taking, perseverance, and leadership, as relevant to an entrepreneur’s success [25]. The authors of [26] classify ten personal skills considered as requirements for an entrepreneurial attitude: looking for opportunities and taking the initiative; accepting risks; having efficiency and quality; showing persistence over time towards a particular goal; showing a high level of motivation; seeking to stay informed; setting measurable, achievable, realistic, specific, defined, and challenging goals; planning and systematically monitoring the actions taken; being persuasive and having support networks; having self-confidence and personal independence; and, finally, committing to oneself or to the projects carried out.

### 1.2. Gender Barriers to Leadership and Entrepreneurship

Nevertheless, more than personal and technical skills are required. There are still many barriers to overcome, especially for women entrepreneurs, despite increasing awareness of the importance of entrepreneurship at different levels. Empirical research evidences a gender gap in business creation that women perceive as an obstacle. However, once the business is running, most women feel equally confident to men about the future, including job creation prospects [27]. The most recent data about the number of women beginning their businesses in 2020 declined much more than the number of men. The pandemic negatively impacted women, overloading them with homeworking and home-schooling, which can explain this trend [22].

Studies approach areas such as reconciling work and family/private life [28], but also areas related to gender differences/gender identity, exposing barriers to entrepreneurship, and especially aspects related to gender differences in occupation/career progression and access to leadership positions [29] and salary differences [30].

To answer the questions “Why is gender relevant?” and “Why can gender still be an obstacle?”, we begin by presenting an excerpt from the Global Entrepreneurship Monitor 2020/2021:

“Inclusiveness in entrepreneurship is critical to any economy, because, if one group in society is not starting businesses on a par with other groups, this will limit job creation, innovation, income generation, the availability of new products and services, and all of the other benefits that new businesses bring to the economy and society. According to this research, in most economies, new businesses are more likely to be started by men than women, although in a few economies the reverse is true and there are others where the gap is small”.

Some authors introduce the equation of “latent entrepreneurship” (measured by the probability of a declared preference for autonomous work concerning employment) to the lack of current entrepreneurship [24], and it stands out that the lack of financial support does not have the explanatory power to justify the magnitude of barriers women have to face. Some practical examples demonstrate that other factors are determinants. For a country such as India, the biggest problem is the lack of a business environment [31], which connects with issues related to marketing, lack of education, health, the involvement of women in family life, and the fact that this is an essentially male-dominated society. In the European case, Halbinsky [27] suggests that women tend to perceive that they lack entrepreneurship skills and have smaller and less effective entrepreneurial networks. Family and tax policies can also discourage female labor market participation, including entrepreneurship. In the European case, other aspects stand out as obstacles to entrepreneurial activity: the bureaucratic issues [32] and cultural and generational features [33]. The maintenance of stereotypes associated with traditional conceptions of gender still constrains women’s role in entrepreneurship. According to these stereotypes, women are characterized by having a personality more associated with family and home care and by more affectionate and helpful traits, while men are, on the contrary, associated with strength, confidence, ambition, and assertiveness—characteristics often used to describe leaders.

The role of leadership can also be perceived as a barrier to female entrepreneurship in several working areas. It implies that gender differences determine the under-representation of women in senior positions. Therefore, difficulty accessing leadership positions for women and their consequent lack of representation have been the subject of several studies [14,34]. In Portugal, the under-representation of women in top management and leadership positions—a phenomenon known as vertical sexual segregation—remains one of the most striking characteristics of Portuguese organizations, as noted by Paço [35]. The metaphor of the glass ceiling can also express this phenomenon. This expression is recognized as “the invisible barrier that prevents women from climbing to the top rung of the corporate ladder, regardless of their qualifications or achievements” [36].

Franco and Selvakumar [31] and Albors Garrigos and colleagues [37] also refer to this trend in women’s careers. Authors such as Brands and Fernandez-Mateo made similar findings while carrying out a study on the effects of rejection on the willingness of women to apply for top management positions. The leadership stereotype negatively triggered uncertainty in the executive domain for women, leading them to be less inclined than men to apply again to a company that rejected them. Possibly, this tendency contributed to a cumulative gender disadvantage and increased gender inequality over time [38]. The fear of failure seems to be another barrier to women’s leadership [39,40,41]. In Spain, this reason—fear of failure—and the perception of the needed entrepreneurial skills and access to formal institutions, e.g., for funding and education, are significant reasons that negatively influence entrepreneurial activity [39].

Women’s access to the entrepreneurial environment, the labor market, and clients is more complex than men’s [42]. This lack of access might lead to the perception of women being less able than men to show initial public offering investments [43] and facing a lack of resources of all kinds [27]. According to the OECD, men-run businesses dominate the export sector [27].

Focusing on work contexts in rural family businesses, Bessière indicates that socialization varies according to gender, with male heirs being privileged in the transmission of professional knowledge and skills and to whom leadership is equally attributed, to the detriment of women [44]. Eib and Siegert conducted a comparative study between France and Germany, trying to understand how family life factors can affect business success. In both countries, it is highlighted that women report working more hours at home and having less success in business than men and that entrepreneurship can only empower autonomous women living alone [28]. Vial [45] focuses on entrepreneurial mothers (mompreneurs). They point out that women who take on the social image of entrepreneurial mothers are mostly upper-class women. Vial explains the gender gap in entrepreneurship in the French institutional context, identifying mothers as innovative entrepreneurs. These authors add that women outperform men in management but do not obtain funding.

Moreover, funding is one of the major obstacles identified in women’s entrepreneurship: “the lack of access to financial services for female entrepreneurs is one of the biggest gender gaps, and a major factor holding back progress towards financial inclusion of women in developing countries” (FMO, Entrepreneurial Development Bank). The European Investment Bank identified a gap in funding for female-led businesses. So, they created the program “Funding Women Entrepreneurs through MFF 2021–2027” to answer this issue. It includes innovation, finance, and advice, taking female-led companies to the mainstream (European Investment Bank, April 2021). Still, data from the OECD and the European Commission on policy factors that guide women in entrepreneurship indicate increased availability of resources for women entrepreneurs, including skills, financial resources, networks, training in entrepreneurship, coaching, and mentoring. One of the critical questions in providing entrepreneurship support to women is whether there is a need to promote specific programs by specialized agencies or whether they can integrate into conventional programs.

Therefore, researchers and businesswomen have identified the maintenance of a gender gap in entrepreneurship and listed the main areas where women feel discriminated against. Departing from this state of the art, we applied a focus group methodology to verify the barriers and psychological factors impacting entrepreneurship and how the required skills for entrepreneurship interact with psychological factors.

## 2. Materials and Methods

### 2.1. Instruments

To fulfil the proposed objectives, we designed two complementary studies. A mixed methodology (quantitative and qualitative) was used, to obtain a better understanding of the problem, and also to compensate for the inherent weaknesses of using each approach separately. The first used a questionnaire based on the scientific literature, designed to obtain quantitative data on entrepreneurs or future entrepreneurs. Second, through the ESTEEM partners, we used the classical focus group methodology [46,47] with different agents and stakeholders related to the entrepreneurship field. Within the qualitative data study, it was decided to use a focus group methodology, instead of individual interviews, to make the process more efficient.

The Global Questionnaire on Psychosocial Factors Affecting Women’s Entrepreneurship (GloPEW) was generated using a Delphi method among the ESTEEM partner teams to evaluate the perceived barriers and other psychosocial factors related to entrepreneurship. First, all the psychosocial variables related to entrepreneurship in the scientific literature were collected in the first phase. Then, in the second phase, a group of psychology, equality, and entrepreneurship experts worked out a preliminary version of the questionnaire, defining the variables and transforming these into clear items.

The GloPEW includes a part divided into different fields related to the type of information obtained. Therefore, the first field is related to socioeconomic data, the second is related to the business support expected or received, and the last is related to the psychosocial profile.

The final version has 24 items about psychosocial variables. The questionnaire included questions such as, “I can always manage to solve difficult problems if I try hard enough”, answered with a Likert scale of five points (1 = Disagree strongly; 2 = Disagree; 3 = Neither agree or disagree; 4 = Agree; 5 = Agree strongly).

As for its reliability, the GloPEW obtained a Cronbach’s alpha higher than 0.80 in every measured variable, a sufficient value to consider it reliable. The final questionnaire version can be consulted in Appendix A.

### 2.2. Variables

After carrying out the desk analysis, literature review, and then the variable selection discussed in a classical three-round Delphi method, there was the selection of the skills and psychosocial variables to be used, which were to confirm the final version of the GloPEW questionnaire. This approximation could not ignore the fact that the relationships between personal predispositions and socio-educational conditions of development present gender biases that largely explain these potential differences.

#### 2.2.1. Entrepreneur Skills

In this study, different skills related to entrepreneurship were evaluated: initiative, leadership, negotiation, networking, communication, creativity, organization, teamwork, self-confidence, commitment to the organization, empathy, flexibility, and customer orientation [48].

#### 2.2.2. Psychosocial Key Variables

*Self-efficacy* is the belief a person has about their ability to perform well and succeed. It is based on different things, such as their previous experiences or the results obtained by other people and the perception of their emotional state [49].

*Psychological well-being* is related to a person’s satisfaction with their life. It consists of different factors: positive relationships with others, personal autonomy, a feeling of purpose and meaning in life, personal growth and development, and the environment domain. This study has focused on the latter, that is, each person’s knowledge and management of their task and environment (excerpted and adapted from [50]).

*Emotional intelligence* is the ability to identify emotions felt, understand and regulate from where they arise and their consequences, and learn to express them appropriately for oneself and others (excerpted and adapted from [51]).

*Perceived barriers* refer to an individual’s assessment of the obstacles to behavior change. For example, even if an individual perceives a healthy behavior as threatening and believes that a particular action will effectively reduce the threat, barriers may prevent engagement in the health-promoting behavior. Overcoming this is closely related to the self-efficacy and emotional intelligence of the person [52,53].

### 2.3. Participants

#### 2.3.1. Study 1

During 2021, we conducted a study through the GloPEW administered online by Google Forms to 296 people, within 4 countries, of which 197 (66.55%) are women and 99 (33.45%) men, with an average age of 39.96 years (SD = 10.14), and of which 141 (44.30%) are future entrepreneurs, 59 (19.90%) are young entrepreneurs (with less than two years of experience), and 96 (35.80%) are well-established entrepreneurs. There are differences between women and men related to academic level, as women have higher academic levels than men. Here, 69 (35.47%) women have tertiary education, and only 17 (17.91%) men have tertiary education.

#### 2.3.2. Study 2

After discussion among the panel of ESTEEM experts, 26 people who voluntarily consented to participate in the study were recruited, following the guidelines on the focus group’s ideal size [54], with the following profiles (Table 2):

### 2.4. Procedure

#### 2.4.1. Study 1

After the necessary adaptations—mostly related to the ESTEEM partner’s language—were carried out, the questionnaire was ready to be answered online through the Google Forms platform and, in a few cases, due to their proximity, through a physical version. We followed an opportunity criterion to recruit participants. We disseminated the questionnaire to the participants through entrepreneurship institutions, public and private educational platforms, and chambers of commerce in different countries. All participants had to complete an online or physical informed consent form before answering the GloPEW. The data collection was carried out from February to March 2021.

#### 2.4.2. Study 2

A focus group (FG) was designed, following the standardized guides for this type of qualitative data collection [54], consisting of four discussion phases carried out online on a Zoom-like platform.

Focus group phases:Presentation of the ESTEEM project, presentation of the results of the quantitative data, and presentation of the participants.Presentation of the topics and open questions about every topic.After a short break, the focus group coordinator analyzed the previous responses of the participants to prepare for the third part, where the participants were asked different questions on topics that needed to be clarified.

In this last phase, participants were asked if they wanted to add something else or if something was unclear before closing the focus group.

### 2.5. Data Analysis

#### 2.5.1. Study 1

A descriptive analysis of the GloPEW answers from entrepreneurs (or future entrepreneurs) was carried out, along with several ANOVAs, including searching for differences between groups (men and women). Furthermore, Pearson correlation analyses were carried out between the socioeconomic data and the psychosocial factors. All these analyses were performed with the statistical software SPSS version 26.

#### 2.5.2. Study 2

Once the focus group had been carried out, each partner transcribed on paper the responses that were analyzed concerning the topics related to the study. Experts identified and recognized verbalizations related to entrepreneurship psychosocial factors with the focus group transcription recordings.

Then, the agreements and disagreements about the participants’ statements were analyzed in a second phase. The aim of analyzing the number of agreements and disagreements was to find if there was a general opinion among the participants.

The last part involved categorizing the quotes into previously designed dimensions, using quantitative analysis. Some were experiences, personal factors, perceived barriers, abilities, expectations, support, or gender differences (Table A2).

## 3. Results

### 3.1. Study 1

In Figure 1, we can observe the means of the responses of the people of our sample. The red lines indicate the theoretical average of the general population in order to remark on the low scores obtained by our sample. All of them are above the theoretical average but only narrowly exceed this. The only factor far exceeding the theoretical average is “entrepreneurship skills”.

In Figure 2, we can observe the means of the responses categorized by gender. The most relevant things when we observe this figure are: (1) men, as in other papers and investigations, have higher scores than women. (2) These differences are not statistically significant. In other words, there are no differences between men and women in the factors evaluated in our sample.

We have been able to observe a high and significant correlation (at 0.01 significance level = **) between the self-efficacy factor and the others (entrepreneurship skills: 0.792 **, emotional intelligence: 0.550 **, environment control: 0.741 **, and perceived barriers: 0.510 **). We did not find any correlation between the academic level and the entrepreneurs’ skills or abilities.

We have observed that the type of business support expected or received is related to these results. People expect more public business support (59.38%) than for private ones (40.63%). Furthermore, we have found that the preferred agents to apply to for business support are different between countries, with the most popular agents being: consultancies (25.97%), chambers of commerce (25.54%), and business accelerators (17.75%). We have found a significant difference between men and women in the willingness to ask for help related to the search for business support. Women are more willing to ask for help than men.

Other findings were related to the reasons why people do not receive or ask for help. Three reasons are the most relevant: “not needing business support” (33%), “lack of information about the type of business support available” (29.63%), and “lack of information about the organizations or institutions which offer that business support” (18.52%).

### 3.2. Study 2

Once the FG had been carried out in the different countries, a table was drawn up and structured according to the previously defined factors to collect the qualitative contributions, analyze them, and compare them with the data obtained in the quantitative study. The participating countries’ main qualitative results of the FGs are shown below.

Table A2 (in Appendix B) shows the different results organized concerning the studied dimensions (experience; personal factors; perceived barriers; skills; business relations and gender differences; support/follow-up; expectations; and esteem) and the agreements and disagreements. Additionally, for better clarification of what happened, there is column of literal quotes, exemplifying verbalizations in this regard.

### 3.3. Comparison of Quantitative and Qualitative Data

GF information was collected and analyzed, comparing it with the quantitative information obtained through the GloPEW, and then it was discussed in a meeting of all the teams. We present the most significant overall results in the following table.

Table 3 shows the differences between the results of the GloPEW quantitative questionnaire and the qualitative results collected in the focus groups.

## 4. Discussion

The results of these studies allow us to shed light on the factors that influence women’s entrepreneurship and the differences with men in this field of entrepreneurship.
First, the results show that two main factors affect the others. One is self-efficacy [55,56,57,58]. This factor correlates highly with other skills and abilities. Furthermore, it reflects the degree to which the person can face challenges successfully, so it is a factor that acts as a possible predictor of entrepreneurship and, above all, of persistence and motivation not to abandon the process. Likewise, continuing to try and make progress also generates an increase in self-confidence, showing a motivational ellipse that makes one advance more and more.Emotional intelligence is the other crucial psychological factor found [59,60,61,62,63]. Again, it is highly statistically significantly correlated with all the other variables because people with greater emotional intelligence are the ones who better manage barriers and have more self-efficacy and greater entrepreneurial skills. Nevertheless, this relationship occurs in all cases, considering age, educational level, gender, or country. Therefore, it is a crucial factor because of its characteristics, which may facilitate the development of favorable psychological conditions for entrepreneurship.It is essential to remark that there are no differences between men and women in any psychosocial variables studied. In other words, women and men have the same perception of low self-efficacy, of their entrepreneurial skills, the same emotional intelligence, and awareness of the barriers they face, regarding what it means to be an entrepreneur. This equality focuses on seeing and resigning oneself to the limitations and difficulties, which means that in addition to the objective limitations that society imposes on women, they also subjectively limit themselves, expecting not to be able to undertake, as happens to most men.In addition, women are more willing to ask for help. Likewise, we must remember the doubts expressed regarding the distribution of time and effort between work and family.It is relevant to show some differences between countries related to the different sociopsychological variables studied [64,65,66]. The descriptive data found indicate that in the groups evaluated in Spain and Italy, there are fewer entrepreneurs with low self-efficacy levels than in France. However, the differences are not statistically significant within and between the other countries. The data obtained from the groups evaluated in France and Portugal indicate that the people of these countries have significantly more entrepreneurial skills than in Italy. However, the rest of the countries have no statistically significant differences. Regarding the data on emotional intelligence and perceived barriers, no statistically significant differences were observed between countries.As can be seen, self-efficacy and emotional intelligence correlate with the other psychosocial variables facilitating their development and improvement. At this point, it is essential to understand that, contrary to the popular belief that the more academic education a person has, the more entrepreneurial skills they have, these results showed that academic level and age are unrelated to better entrepreneurship abilities or skills and that any age or academic level is suitable for undertaking projects.Finally, one of the most critical barriers perceived is professional support for entrepreneurship. Almost half of the people evaluated say they need more information about the type of support available and/or about the organizations that offer it. In this last aspect, there are differences by country. They rely mainly on consultancies in Spain and Italy, but chambers of commerce in France. In Portugal, consultancies are also chosen, but to a much lesser extent and with firms for companies. Only a tiny percentage of evaluated people indicated that they do not need professional support for entrepreneurship.

In short, different psychological factors have to be considered to improve women’s entrepreneurship—factors such as self-efficacy or emotional intelligence act as catalysts for the remaining psychosocial variables. In addition, although there are no significant differences between men and women in the different psychosocial variables, there are differences by country. These differences between countries highlight the importance of delving into the cultural idiosyncrasy of each country when adopting specific measures. Furthermore, these differences between countries are seen mainly in terms of business support. Therefore, the measures adopted on this point may differ between countries. Additionally, although the data show statistical strength, it seems necessary to expand the sample and incorporate more profiles of female entrepreneurs, for example, with different levels of training and combinations of characteristics (for example, age) given the complexity and variety of intervening factors.

It is a relevant fact that almost half of the sample is unaware of the types of entrepreneurial aid that exist. That is why it is so significant to provide targeted information, such as preparing specific training courses, based on the data obtained in this study.

### Practical Applications and Further Developments

This study makes three main contributions to knowledge about female entrepreneurship. First of all, it points out that, although there are several factors that affect it, some of them are more relevant (self-efficacy and emotional intelligence); secondly, there are factors such as cultural differences and the socio-labor organization of the specific countries that require identification; and, thirdly, it has been possible to build an evaluation instrument, with adequate guarantees of reliability and validity. Likewise, it may serve to develop intervention strategies, also to be taken into account.

The results provided a straightforward guide to elaborating and implementing psycho-educative courses. The pillars of these courses should be: the enhancement of self-efficacy; the use of emotional intelligence; the discovery and way of overcoming the perceived barriers to entrepreneurship; and the control of the environment (general or specific), addressed to improve the efficacy and well-being of women trying to initiate or maintain entrepreneurship.

These psychosocial courses should teach some of the skills (using questionnaires such as the GloPEW to find the gaps in a given population or persons) which have been found relevant to women’s entrepreneurship development. On the other hand, it would be interesting for future studies to analyze the participants’ need for more information on the existing business support programs.

Although there are differences between the countries and even among different regions of each country, it is possible to design general skilling programs supported by European or national public policies, requiring just some adaptation to each economic environment.

## Figures and Tables

**Figure 1 behavsci-13-00313-f001:**
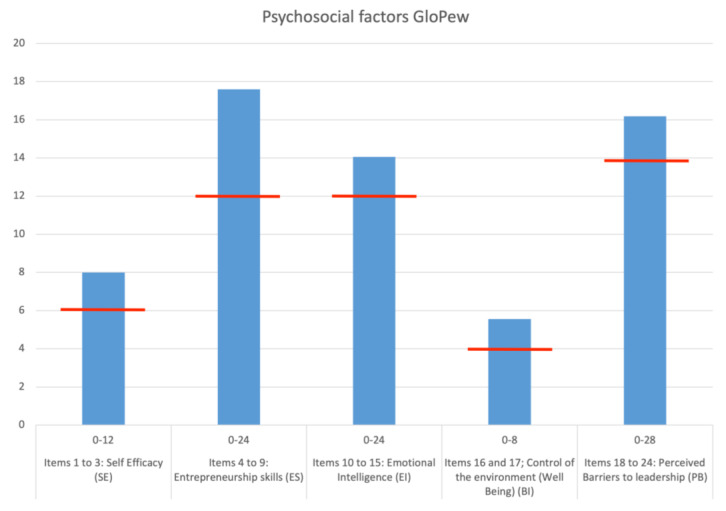
Mean values of the GloPEW biopsychosocial factors.

**Figure 2 behavsci-13-00313-f002:**
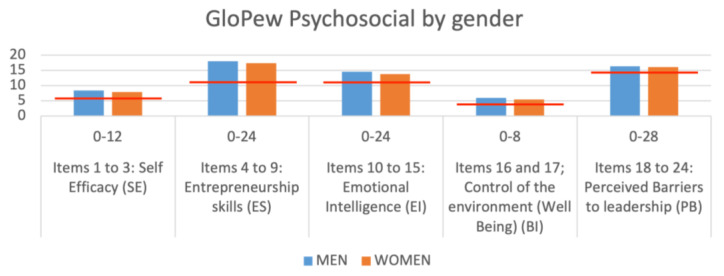
Mean values of the GloPEW psychosocial factors by gender.

**Table 1 behavsci-13-00313-t001:** Classification of female entrepreneurship (FE).

	Individual	Collective
Social	FE is the same concept as entrepreneurship	FE substitutes the individual concept and its characteristics can be had by male entrepreneurs
Natural	FE is an homothetic concept with respect to entrepreneurship	FE substitutes the individual concept and its characteristics cannot be had by male entrepreneurs

**Table 2 behavsci-13-00313-t002:** Recruited participants per country.

Country	Participants	Age	Business Area	Previous Experience in the Business Area	Time/Type of Experience in Entrepreneurship
Spain	5 (3 men and 2 women)	30–50	Business advice; Creative architecture; Social education; Private high school teacher; Industrial engineer	Yes (5)	15 years (1); 6 years (1); 10 years (1); No experience (2)
Portugal	6 (3 women and 3 men)	29–47	Skin cosmetics; Digital technologies; Communication marketing; Video production and editing; Industrial design, architecture and visualization; Tourism (restaurant business)	Yes (5) and No (1)	Young (3) and Well-established (3)
Italy	8 (3 women and 5 men)	29–44	Photography; Dress maker; Private school; Services (food); Social enterprise; Agriculture sector; Cultural organization; Bar/Restaurant	n/a (8)	Future entrepreneur (3); 15 years (3); 8 years (1); 20 years (1);
France	7 women	23–57	furniture manufacturing; Cosmetic manufacturing; Bakery; Manufacturing of lingerie (underwear); Sewing (2); Ceramics	Degree (4); None (3)	3 years (2); 2 years (2); 6 months; 3 months; No experience

**Table 3 behavsci-13-00313-t003:** Comparison of main quantitative and qualitative results.

GloPEW	FGs
There are differences between countries when requesting business support	Lack of information about support and financial aims
There are no differences between men and women in biopsychosocial variables (self-efficacy, entrepreneurship skills, emotional intelligence, control of the environment, and perceived barriers to leadership)	
Only a small percentage (33%) indicate that they do not need professional support for entrepreneurship	Lack of information about support and financial aims
Self-efficacy correlates highly with the other skills and abilities	Self-efficacy shows high agreement (between FGs)
The academic level is not related to entrepreneurship abilities or skills	Social skills (communication, negotiation) Resilience (self-confidence, e.g.: “I know that I’ll finally get it”) Problem solving Leadership Adaptability (flexibility and ability to adapt to changes, e.g.: “You have to find new ways of doing things when circumstances change”) Teamwork Grit (courage/tolerance to the risk) (attitude: be willing to do things in a different and novel way) Technical skills (related to the activity) (attitude: open to continuing to learn new things) (they are human abilities not directly related to the level of academic training of the person)
Women are more willing to ask for help	
Perceived barriers	Gender roles in labor market (age of women is a specific factor)

## Data Availability

All collected data is contained within the article, and presented in the Tables, there is no other data collected.

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
