# Peer review of "Quantitative and Qualitative Analysis of Psychosocial Factors Affecting Women’s Entrepreneurship"

_behavsci, 2023, doi:10.3390/bs13040313_

Round 1
Reviewer 1 Report
Overall the paper is well thought off. However, there are some issues that needs addressing before acceptance. I will highlight these issue for further improvement.
1) Literature review/background- while 3 focus areas have been mentioned, the paper suffers from a clear articulation of a research aim. What is the main research area/question that the authors want to address?
2) Methods: This is a strong component in the paper, however, some of the tables and figures are not in English. A better explanation of how the quants and qualitative approach helps in answering the questions is needed. for example, why focus group?
3) Contribution: Please highlight the contribution to theory more explicitly, at the moment, this is not coming out clearly
Author Response
1) Literature review/background- while 3 focus areas have been mentioned, the paper suffers from a clear articulation of a research aim. What is the main research area/question that the authors want to address?
We have added a paragraph in the introduction to clarify this point. (lines 113-117)
2) Methods: This is a strong component in the paper, however, some of the tables and figures are not in English. A better explanation of how the quants and qualitative approach helps in answering the questions is needed. for example, why focus group?
In the Methodology section, an explanatory paragraph has been added on why to use a mixed methodology. (lines 242-244). In addition, the FG's decision is explained (lines 248-250).
Regarding the tables. Some of them have been removed to integrate the results into the text. Figure 3: Correlations - removed. Instead, significant correlations have been integrated into the text (lines 373-378).
Figure 4: Business distribution- removed. Instead, the data has been integrated into the text. (lines 379-380).
Figure 5: Agents preferred- Removed. Instead, the main findings have been integrated into the text (lines 379-34).
Figure 6: Reasons to not ask for help- Removed. Instead, the main results have been integrated into the text. (lines 388-392).
3) Contribution: Please highlight the contribution to theory more explicitly, at the moment, this is not coming out clearly
We have added a paragraph in the Discussion explaining the main contributions (477-483).
Reviewer 2 Report
The paper presents an interesting mixed methodology design and addresses certain gaps in the literature in reference to women entrepreneurs. However, I found it very vague in the findings and discussions of the results with colored graphs and copy paste of statistical analysis from excel, which leaves enormous room for improvement.
Author Response
The paper presents an interesting mixed methodology design and addresses certain gaps in the literature in reference to women entrepreneurs. However, I found it very vague in the findings and discussions of the results with colored graphs and copy paste of statistical analysis from excel, which leaves enormous room for improvement.
In response to the request, it has been considered better to eliminate figures: 3,4,5 and 6. and integrate their content into the text.
Figure 3: Correlations - removed. Instead, significant correlations have been integrated into the text (lines 373-378).
Figure 4: Business distribution- removed. Instead, the data has been integrated into the text. (lines 379-380).
Figure 5: Agents preferred- Removed. Instead, the main findings have been integrated into the text (lines 379-34).
Figure 6: Reasons to not ask for help- Removed. Instead, the main results have been integrated into the text. (lines 388-392).
Regarding the discussion and conclusions:
We have added a paragraph in the Discussion explaining the main contributions (477-483).
Reviewer 3 Report
Recommendations for the authors of the article:
1. The "abstract" section of the article should be corrected. It lacks, among other things, a description of the methodological issues of the article, including research limitations.
2. Add a section: "Literature review". Particular attention should be paid to the conditions for the development of entrepreneurial qualities of women due to their personal predispositions.
3. In the article the conclusions of the studies should be given in sub-paragraphs. In addition, there is no description of the research limitations.
4. Finally, it is necessary to emphasize the proposed and adopted directions of actions for strengthening social capital in the studied population.
Author Response
- The "abstract" section of the article should be corrected. It lacks, among other things, a description of the methodological issues of the article, including research limitations.
Using a mixed methodology to compensate for the inherent weaknesses of using each approach. // Although the data shows statistical strength, it seems necessary to expand the sample and incorporate more profiles of female entrepreneurs, for example, with different levels of training, given the complexity and variety of intervening factors. (line 11 // line 16)
- Add a section: "Literature review". Particular attention should be paid to the conditions for the development of entrepreneurial qualities of women due to their personal predispositions.
This approximation should not ignore the fact that the relationships between personal predispositions and socio-educational conditions of development present gender biases that largely explain these potential differences. (line 272)
- In the article the conclusions of the studies should be given in sub-paragraphs. In addition, there is no description of the research limitations.
Also, although the data shows statistical strength, it seems necessary to expand the sample and incorporate more profiles of female entrepreneurs, for example, with different levels of training, and combinations of characteristics (for example age and training) given the complexity and variety of intervening factors. (line 491)
- Finally, it is necessary to emphasize the proposed and adopted directions of actions for strengthening social capital in the studied population.
It has been answered before.
Round 2
Reviewer 3 Report
Dear Authors, I think in this version the article is scientifically, methodologically and empirically on a good level. Congratulations. I wish you scientific and professional success.